# COST-EFFECTIVE TESTING OF A DEEP LEARNING MODEL THROUGH INPUT REDUCTION

## ABSTRACT

With the increasing adoption of Deep Learning (DL) models in various applications, testing DL models is vitally important. However, testing DL models is costly and expensive, especially when developers explore alternative designs of DL models and tune the hyperparameters. To reduce testing cost, we propose to use only a selected subset of testing data, which is small but representative enough for quick estimation of the performance of DL models. Our approach, called DeepReduce, adopts a two-phase strategy. At first, our approach selects testing data for the purpose of satisfying testing adequacy. Then, it selects more testing data in order to approximate the distribution between the whole testing data and the selected data leveraging relative entropy minimization. Experiments with various DL models and datasets show that our approach can reduce the whole testing data to 4.6% on average, and can reliably estimate the performance of DL models. Our approach significantly outperforms the random approach, and is more stable and reliable than the state-of-the-art approach.

## 1 INTRODUCTION

In recent years, deep learning (DL) models have been successfully deployed in a variety of application domains. Erroneous DL models may lead to severe consequences. For example, a defect in a DL-based autonomous driving system resulted in the death of one pedestrian (Amir, 2018). Therefore, sufficient testing for DL models is critical for ensuring system quality. However, the cost for testing deep learning models is not neglectable. For example, DeepFace (Taigman et al., 2014), the face recognition system of Facebook, used about 0.22 million face images for testing. Tian et al. (2018) generated 254,221 images for testing a Chauffeur-CNN based autonomous driving model and achieved a neuron coverage (the ratio of activated neurons) of 88%. The popular ImageNet dataset (Russakovsky et al., 2015) contains 100,000 testing images (100 per class) for testing various image classification models. Clearly, it could take a long time to sufficiently test a DL model. Furthermore, in practice, developers always need to evaluate alternative designs of DL models and optimize the models. For example, they need to explore different deep neural network (DNN) structures (such as adding/deleting a layer) or different hyperparameters (such as learning rates and dropout rates), which all lead to increased testing cost.

To reduce the testing cost, it is desirable to have an early estimation of the performance of a DL model by using only a small amount of testing data. After the first end-to-end system is established, developers can quickly test the performance of the DL model using a small subset of testing data, and start evaluating the design of the model. If the performance is acceptable, the developers can perform a full-scale testing. Otherwise, the developers can seek alternative designs of the model (e.g., adjust DNN structures or learning algorithms), or further tune the hyper-parameters (such as the learning rate). In this way, developers can quickly obtain an accurate DL model without having to perform costly full-scale testing many times.

To achieve an early estimation of the performance of a DL model, the key challenge is to *select a small yet effective subset of testing data* for testing the model. The selected data should be able to achieve similar testing performance as what the entire testing data achieves, and should be small enough to reduce the testing cost. Random sampling is not an ideal solution as it tends to select more testing data in order to represent the whole testing data. Li et al. (2019) recently proposed a method to reduce the number of unlabelled instances required by DNN testing through a carefully

designed sampling strategy. However, they require users to specify the number of data instances as the input to their approach. That is, without a proper value of such an input, they have no guarantee that the selected data is sufficient for DNN testing. Furthermore, being based on random sampling, their results could be unstable in different runs.

In this paper, we propose DeepReduce, a technique for cost-effective estimation of the performance of a DL model. DeepReduce can significantly reduce the amount of testing data for testing a DL model but can still achieve comparable performance as the whole testing data achieves. We formulate the input reduction problem in this work as a multi-objective optimization problem: 1) *minimize* the amount of testing data selected from the whole testing data; 2) *maximize* the testing adequacy achieved by the selected testing data; and 3) *maximize* the similarity of output distributions achieved by the selected and the whole testing data. These objectives take into account the efficiency, completeness, and effectiveness of testing in order to make the performance estimation more reliable. Inspired by the usage of structural coverage in conventional software testing, we use neuron coverage of a DL model to measure its testing adequacy in this work. We use the outputs of the neurons in the last hidden layer of a DL model to represent the output distribution. In order to maximize the similarity between the output distribution achieved by the selected data and the whole testing data, we design a heuristic-based algorithm to guide relative-entropy minimization.

To evaluate our approach, we conduct an extensive experiment on six DL models and two datasets. The experiment results show that DeepReduce can reduce the amount of testing data required for a reliable testing from 10,000 to 455 on average, which means that over 95.5% testing cost can be saved by DeepReduce. Compared with the state-of-the-art approach (Li et al., 2019), DeepReduce is more stable and reliable, and can reduce more testing data. The experiment results also shows that DeepReduce can be used in performance estimation in regression scenarios.

In summary, we make the following contributions in this work:

- A novel approach to cost-effective testing of a DL model by considering three different objectives: efficiency, completeness, and effectiveness.

- An extensive empirical evaluation to investigate the performance of our approach on real-world DL models. The results have confirmed that the proposed approach can significantly reduce DL testing cost by reducing the amount of input data required for testing a DL model.

## 2 RELATED WORK

**Cost of deep learning**: Achieving the highest accuracy has been the ultimate goal of many DL applications. In recent years, the cost of a DL model has also received considerable attentions. For example, Canziani et al. (2016) described a comprehensive analysis of important metrics in practical DL applications such as memory footprint, parameters, inference time and power consumption. Many techniques have also been proposed to reduce DL cost in practice. For example, Karpathy & Fei-Fei (2015) improved the testing time of image description generation from 3,170ms to 240ms for each image by designing a new network architecture. Ning et al. (2019) proposed Adaptive Deep Reuse, which can reduce training time for deep learning networks by more than 60 percent without sacrificing accuracy. The cost of constructing a DL model could also be reduced through pretrained models and transfer learning (Glorot et al., 2011; Thrun, 1996; Yosinski et al., 2014). In our work, we propose a software engineering approach to reduce testing cost in DL practice.

**Deep learning testing**: Successful applications of deep learning techniques require thorough testing of the DL models. Existing research mainly focuses on generating adversarial examples (Pei et al., 2017; Tian et al., 2018; Zhang et al., 2018) and proposing new coverage criteria (Pei et al., 2017; Ma et al., 2018; Sun et al., 2019) for testing DNNs. However, they ignore the cost of DNN testing, which is also an important problem and cannot be ignored. The costs of DNN testing mainly lie on manual labeling and regression testing. For the former, labeling for those unlabeled data is usually conducted by human, and the costs for labeling is high. Therefore, it is desirable to improve testing efficiency by using a reduced subset of testing data.

In general, the input reduction problem can be viewed as an instance of the existing test minimization problem (Harrold et al., 1993; Yoo & Harman, 2012) in conventional software testing, which aims

to find the smallest subset of tests satisfying the same test requirements (e.g., structural coverage). Previous work pointed that test reduction is an NP-complete problem (Yoo & Harman, 2012), and thus the input reduction problem is also NP-complete. Although a large number of test minimization techniques have been proposed in the literature, none of them can be applied directly to solve the input reduction problem, because the former techniques usually focus on structural coverage alone.

Recently, Li et al. (2019) proposed a method to reduce the amount of unlabelled testing data required in operational DNN testing through a carefully designed sampling for the purpose of reducing the cost of manual labelling. In particular, this approach iteratively constructs several groups with un-labelled data randomly picked from all unlabeled data, and then enlarges the selected set with the group that contributes the most to cross-entropy minimization. However, this approach requires users to specify the amount of selected data instances as input, and cannot guarantee that testing is sufficient. Our approach addresses this problem in two ways: 1) ensuring testing adequacy achieved by the selected testing data is the same as the whole testing data; 2) approximating the output distri-bution between the selected and the whole testing data without the need for specifying the number of data instances.

## 3 PROPOSED APPROACH

### 3.1 PROBLEM DESCRIPTION

In order to estimate the performance of a DL model in a cost-effective way, this paper attempts to select a representative subset of testing data from any given set of testing data, which need labelling and are subsequently used for testing process. Considering the usage of the selected data, i.e., estimating the performance of a DL model, they are expected to satisfy at least two objectives. The first one is that the selected data are expected to have the same testing adequacy (e.g., structural coverage) as the given set, so as to guarantee that the performance estimation is conducted on the complete learning model, instead of partial learning model. The second one is that the selected data are expected to generate the same output distribution as the given set, so as to guarantee the estimation is correct and precise. In other words, the problem this paper targets can be viewed as an input reduction problem, which is formally defined as below.

**Definition 1.** *Given a set of testing data $T$ and its target DL model $M$, we define function $f(T, M)$ to measure to what extent the testing data $T$ cover the structure of $M$ and function $g(T, M)$ to measure the output distribution of $M$ with $T$. The problem of input reduction in DL testing is to find the smallest subset of testing data from $T$, which is denoted as $T'$, satisfying that $f(T, M) = f(T', M)$ and $g(T, M) = g(T', M)$.*

As the input reduction problem is NP-complete, we tune its definition by relaxing its requirements on the output distribution $g(T', M)$. That is, we replace $g(T, M) = g(T', M)$ by $g(T, M) \simeq g(T', M)$, which refers to the output distribution $g(T, M)$ is similar to $g(T', M)$. Such a require-ment on output distribution guarantees that the estimation using $T'$ is close to or even the same as $T$ on $M$.

To solve the redefined input reduction problem, we present a new technique DeepReduce to select a small set of testing data for cost-effective testing. In particular, we first present the metrics used in DeepReduce to define functions $f$ and $g$ (in Section 3.2), and then a two-phase reduction algorithm used in DeepReduce to select testing data (in Section 3.3).

### 3.2 INPUT REDUCTION METRICS

Structural coverage is widely used in conventional software testing, which measures to what extent structural elements (e.g., statements, methods, functions) are covered by a test or test suite. There-fore, structural coverage is usually taken as the default testing requirement of test minimization, so is in input reduction. In the literature, there are various structural coverage criteria (e.g., Neuron Coverage (Pei et al., 2017), k-Multisection Neuron Coverage, and Neuron Boundary Coverage (Ma et al., 2018)) proposed for DL models, each of which can be used to define the function $f$. In this work, we use Neuron Coverage (abbreviated as NC) as a representative structural coverage criterion since it is lightweight. NC (Pei et al., 2017) is defined as the ratio of activated neurons of a DL model, where an activated neuron refers to the neuron whose output of an input data is larger than a

given coverage threshold $\beta$. In particular, NC with various coverage threshold $\beta$ is taken as different coverage criteria (Pei et al., 2017). Following the previous work (Pei et al., 2017), we use NC$\beta$ to represent NC with various coverage threshold $\beta$. Note that our approach is not specific to these criteria, and we will investigate the proposed input-reduction approach by using other structural coverage criterion in the future.

In this paper, we measure the data distribution by leveraging the outputs of the neurons in the last hidden layer. In particular, given a set of testing data $T$ and its target DL model $M$ (which consists of $m$ neurons in its last hidden layer), we record the output range of $T$ on each neuron $e_i$ ($i \in [1, m]$) and divide each output range into $K$ equal sections $D_{e_i,1}, D_{e_i,2}, ...,$ and $D_{e_i,K}$. In other words, the output of each testing data $t$ ($t \in T$) on $e_i$ falls into one of these sections. Therefore, for each $e_i$ in the last hidden layer of $M$, we calculate the percentage of testing data whose corresponding output falls into $D_{e_i,1}, D_{e_i,2}, ...,$ and $D_{e_i,K}$, and then denote the corresponding results by $P_T^{e_i} = \{P_T^{e_i}(1), ..., P_T^{e_i}(K)\}$, where $P_T^{e_i}(j)$ denotes the percentage of input data in $T$ whose corresponding output falls into $D_{e_i,j}$. Finally, the distribution of the given model $M$ with $T$ is represented as $P_T : \{P_T^{e_1}, P_T^{e_2}, ..., P_T^{e_m}\}$.

### 3.3 Two-phase Reduction Algorithm

With the structural coverage $f$ and output distribution $g$ mentioned in Section 3.2, we propose a two-phase reduction algorithm, which is called DeepReduce, with the purpose of selecting testing data by satisfying $f(T, M) = f(T', M)$ and $g(T, M) \simeq g(T', M)$. In particular, our two-phase reduction algorithm first selects a set of testing data to guarantee the structural coverage by reusing the existing HGS algorithm (Harrold et al., 1993), and then selects more testing data with the purpose of maximizing the similarity of output distribution between $T$ and $T'$.

For a learning model $M$ with the given testing data set $T$, we use $P_T$ to represent its output distribution, which is calculated by the ratio of testing data falling into sections $D_{e_i,j}(i \in [1, m], j \in [1, K])$ following Section 3.2. For the selected testing data set $T'$, we calculate its output distribution on the sections $D_{e_i,j}(i \in [1, m], j \in [1, K])$ produced by $T$ instead of $T'$ for ease of comparison, which is denoted by $P_{T'}$. We use relative entropy (Kullback & Leibler, 1951)[1], which is also called Kullback-Leibler Divergence, to measure the similarity between them (denoted as $KL(P_T, P_{T'})$). The calculation is defined in Formula 1.

$$KL(P_T, P_{T'}) = \frac{\sum_{i=1}^{m} \sum_{j=1}^{K} P_T^{e_i}(j) * \log P_T^{e_i}(j)/P_{T'}^{e_i}(j)}{m} \tag{1}$$

Relative entropy is used to measure how one output distribution is different from the other one. Note that the output distribution of $T$ or $T'$ contains $m$ probability distributions, we take their average as the similarity between two output distributions[2]. That is, the smaller the value is, the more similar these two output distributions are.

Algorithm 1 presents the overall framework of our two-phase reduction algorithm, i.e., DeepReduce. In particular, given a testing data set $T$ and its learning model $M$, DeepReduce aims to produce a reduced subset of testing data by satisfying that (1) neuron coverage of $T'$ is the same as that of $T$ and (2) small relative entropy value between $T$ and $T'$, indicating the similarity between the output distribution of $T$ and $T'$ is high. In particular, we use $\alpha$ to represent the threshold of relative entropy, which is usually set to be a very small value like 0.005 in practice. More discussion on the choice of this value and its influence can be found in Section 4. Besides, to facilitate the selection of testing data, DeepReduce also records the following information before applying it. $Cov$ records the NC coverage of $T$, $TD$ is a set recording the output distribution of $T$ (i.e., $P_T$ calculated following Section 3.2), and $TI$ is an array, in which each element records the sections of neurons that the corresponding testing data falls into. $TI$ is then used in Algorithm 2, and more details are describe later in this section.

---

[1] Note that we do not use cross entropy as previous work (Li et al., 2019), because in the special case that $T$ is equal to $T'$, their cross-entropy values on some DL models may differ to a large extent.

[2] The average result can be used to represent the similarity between output distribution, because a DL model typically reduces the correlation among the neurons in its last hidden layer (Koller & Friedman, 2009).

In the first phase, DeepReduce reuses the HGS (Harrold et al., 1993) algorithm to select the minimized subset of $T$ with the same NC coverage. In particular, the HGS algorithm is a greedy algorithm which tends to select testing data covering the activated neurons that are less covered by the existing testing data. In Line 2 of this algorithm, DeepReduce uses $HGS(T, Cov)$ to represent the output of the HGS algorithm on $T$ by satisfying the NC coverage recorded by $Cov$, which is a subset of selected testing data. In Line 3, DeepReduce uses a function $\text{Update}(TD, T')$ to calculate the output distribution of the selected data set $T'$ (denoted as $TD'$) by using the section division generated by $TD$, to facilitate the output comparison between $T$ and $T'$ according to Formula 1. In Line 4, DeepReduce put the remaining unselected testing data into $REMAIN$, which are to be selected in the second phase. In the second phase, DeepReduce selects more testing data (according to Lines 6-11) until the set of selected data is similar to $T$ in output distribution. That is, more testing data are selected until the termination criterion (i.e., the relative entropy value between $T$ and $T'$ is no larger than the specified threshold $\alpha$) is satisfied, shown by Line 6.

---

**Algorithm 1:** Two-Phase Reduction

**Input** : Input data: $T$; model: $\mathcal{M}$, coverage: $Cov$, output distribution of $T$: $TD$, output section: $TI$, and the threshold of relative entropy: $\alpha$

**Output:** Reduced test set $T'$

1  // *First Phase*;
2  $T' \leftarrow HGS(T, Cov)$;
3  $TD' \leftarrow \text{Update}(TD, T')$;
4  $REMAIN \leftarrow T \setminus T'$;
5  // *Second Phase*;
6  **while** *KL(TD, TD')* $\geq \alpha$ **do**
7  $\quad t' = \text{getCandidate}()$;
8  $\quad T' \leftarrow T' \cup \{t'\}$;
9  $\quad REMAIN \leftarrow REMAIN \setminus \{t'\}$;
10 $\quad TD' \leftarrow \text{Update}(TD, T')$;
11 **end**
12 **return** $T'$;

---

To facilitate the selection of testing data to minimize the relative entropy value between $T$ and selected data, DeepReduce designs a function getCandidate() to iteratively select testing data from $REMAIN$, shown by Algorithm 2. In Lines 2-11, DeepReduce seeks to find the section in which the output distribution differs the most between $T'$ and $T$ on each neuron, and these sections are recorded based on the order of the corresponding neuron by $re$. In particular, for each neuron $e_i$, DeepReduce calculates the output difference on each section $D_{e_i,k}$ between $T$ and $T'$ through $TD_{i,k}/TD'_{i,k}$, so as to find the section with the largest output difference on each neuron (denoted as $kmax$ in the algorithm). Note that in this section selection process, DeepReduce uses $TD_{i,k}/TD'_{i,k}$ instead of $P_T^{e_i}(j) * \log P_T^{e_i}(j)/P_{T'}^{e_i}(j)$ (defined in Formula 1) because they produce almost the same selection results and the former reduces the computation cost to a large extent. Then Lines 12-26 are to select the testing data $t'$ that may minimize the output difference between $T$ and the selected testing data set. As variable $re$ represents the output distribution difference between $T$ and $T'$, the testing data whose output distribution is similar to $re$ (which is measured by the getSimilarity function) are expected to be selected so as to minimize the output distribution difference between $T$ and the selected testing data set. Following this intuition, in Lines 13-20, for each $t$ in $REMAIN$, DeepReduce iteratively calculates the number of neurons in $TI[t]$ and $re$ with the same section, and finds the testing data with the largest number of such neurons. $dict$ is an array recording candidate

---

**Algorithm 2:** getCandidate()

**Input** : $TD'$, $TI$, $T'$, $REMAIN$, $m$, $K$

**Output:** Candidate test : $t'$

1  $re \leftarrow \emptyset$;
2  **foreach** $i \in \{1, m\}$ **do**
3  $\quad kmax \leftarrow 1$;
4  $\quad maxvalue \leftarrow TD_{i,1}/TD'_{i,1}$;
5  $\quad$ **foreach** $k \in \{2, K\}$ **do**
6  $\quad\quad$ **if** $maxvalue < TD_{i,k}/TD'_{i,k}$ **then**
7  $\quad\quad\quad kmax \leftarrow k$;
8  $\quad\quad\quad maxvalue \leftarrow TD_{i,k}/TD'_{i,k}$;
9  $\quad$ **end**
10 $\quad re \leftarrow re \cup \{kmax\}$;
11 **end**
12 $maxsim \leftarrow 0, dict \leftarrow \emptyset$;
13 **foreach** $t \in REMAIN$ **do**
14 $\quad sim \leftarrow \text{getSimilarity}(TI[t], re)$;
15 $\quad$ **if** $sim > maxsim$ **then**
16 $\quad\quad maxsim \leftarrow sim$;
17 $\quad\quad dict \leftarrow \{t\}$;
18 $\quad$ **else if** $sim == maxsim$ **then**
19 $\quad\quad dict \leftarrow dict \cup \{t\}$;
20 **end**
21 $minvalue \leftarrow \infty$;
22 **foreach** $t \in dict$ **do**
23 $\quad$ **if** $minvalue > KL(T, T' \cup \{t\})$ **then**
24 $\quad\quad minvalue \leftarrow KL(T, T' \cup \{t\})$;
25 $\quad\quad t' \leftarrow t$;
26 **end**
27 **return** $t'$;

testing data, each of which has the largest similarity with $re$. Among all the candidates in $dict$, in Lines 21-26, DeepReduce determines which testing data should be chosen by iteratively calculating the relative entropy between the given testing data set $T$ and the selected testing data set by including each candidate.

# 4 EXPERIMENTS

In this section, we present an experimental study to investigate the performance of the proposed approach from three aspects:

- RQ1: How does our approach perform in reducing testing data?
- RQ2: How do different parameters (i.e., coverage criteria and termination criteria) influence our approach?
- RQ3: How does our reduction approach perform in regression scenario?

## 4.1 DATASETS

In this work, we consider two image recognition datasets, MNIST and CIFAR-10. MNIST (LeCun et al., 1998) is a typical dataset widely used in machine learning research. It contains 60,000 training images and 10,000 testing images in total. We train three LeNet family models (LeNet-1, LeNet-4, and LeNet-5) (LeCun et al., 1998) and evaluate our approach on them. CIFAR-10 (Krizhevsky et al., 2014) is another widely used dataset, which contains 50,000 training images and 10,000 testing images. On this dataset we train three DL models, including NIN (Lin et al., 2013), VGG19 (Simonyan & Zisserman, 2014), and ResNet (He et al., 2016). Table 1 shows the detailed information of these models along with their test accuracy obtained by the whole testing data.

| Metric | CIFAR-10 | | | MNIST | | |
|---|---|---|---|---|---|---|
| | NIN | VGG19 | ResNet | LeNet1 | LeNet4 | LeNet5 |
| Neurons | 3,432 | 50,782 | 4,138 | 52 | 148 | 268 |
| Layers | 24 | 65 | 113 | 7 | 8 | 9 |
| Accuracy | 0.8815 | 0.9346 | 0.9220 | 0.9857 | 0.9888 | 0.9884 |

Table 1: Subject information

## 4.2 EXPERIMENTAL DESIGN

For RQ1, we first collect neuron coverage and output distribution of the testing data from the trained models, and then apply our approach on the whole testing data to obtain a subset. In order to investigate the effectiveness of our approach (DeepReduce), we compare it with both random approach and BOT (Li et al., 2019). BOT is the state-of-the-art approach in DL testing. Different from DeepReduce, BOT requires to specify the number of data instances to be selected. Besides, it does not consider satisfying structural coverage in data selection. To enable comparison, we implement a variant of BOT, $BOT_v$, by changing the termination criteria to the one used in DeepReduce. That is, $BOT_v$ iteratively selects testing data until the relative entropy value is smaller than $\alpha$. We also implement a random reduction approach, which randomly selects testing data from the whole testing data until the relative entropy value is smaller than $\alpha$. Both $BOT_v$ and random approach are repeated 50 times to reduce the influence of their inherent randomness, and the average values are used for comparison.

For RQ2, we study the influence of two parameters (i.e., coverage criteria and termination criteria) on our approach. The neuron coverage can be changed when different activation thresholds are given. The termination criteria (i.e., the relative entropy value between the output distributions of the selected data and the whole data) also affects the amount of selected data. Thus, we evaluate our reduction approach with different settings in RQ2.

For RQ3, we evaluate our approach on various modified models with the purpose of investigating the effectiveness of our reduction approach in regression scenario. We first modify the original DL models and retrain the models in order to simulate regression scenario. Then, we evaluate our approach on the modified models by comparing the accuracy of the selected data with that of the whole testing data. We expect that the performance of the modified DL models can still be estimated by using the data selected by DeepReduce. In this work, we apply seven changes in total for all

the studied models, including adding layer (ADL), deleting layer (DEL), adding neurons (ADN), deleting neurons (DEN), changing learning rate (LR), changing momentum (MO), and changing dropout ratio (DR). Although these changes do not cover all the cases where a DL model is modified in regression scenarios, they are typical changes and cover both structural (related to model structure) and non-structural (unrelated to model structure) changes.

## 4.3 EXPERIMENTAL RESULTS

### 4.3.1 RESULTS FOR RQ1

In this RQ, we investigate the effectiveness of DeepReduce in reducing testing data for estimating the performance of a DL model. We take the default setting of two parameters, i.e., threshold for neuron coverage and $\alpha$. For the former, we take the recommended value (0.5) given by Pei et al. (2017). For the latter, we assign 0.005 to $\alpha$ when the termination criteria is used in the algorithm. Table 2 presents the results for comparison. In the table, "Acc." represents the test accuracy achieved by the selected data, and "Size" represents the amount of selected data.

| Approach | Metrics | CIFAR-10 | | | MNIST | | |
|---|---|---|---|---|---|---|---|
| | | NIN (0.8815) | VGG19 (0.9346) | ResNet (0.9220) | LeNet1 (0.9857) | LeNet4 (0.9888) | LeNet5 (0.9884) |
| DeepReduce | Acc. | 0.8316 | 0.9382 | 0.8779 | 0.9757 | 0.9923 | 0.9957 |
| | Size | 487 | 552 | 827 | 371 | 261 | 233 |
| Random | Acc. | 0.8826 | 0.9345 | 0.9221 | 0.9860 | 0.9886 | 0.9907 |
| | Size | 3398 | 3258 | 4903 | 3159 | 2436 | 2589 |
| $BOT_v$ | Acc. | 0.8571 | 0.9335 | 0.8959 | 0.9854 | 0.9905 | 0.9908 |
| | Size | 557 | 438 | 835 | 381 | 337 | 337 |

Table 2: The effectiveness of DeepReduce in input reduction

From the table, we find that DeepReduce can reduce the whole testing data significantly, ranging from 261 (2.6%) to 827 (8.3%), with an average of 455 (4.5%). Also, we can find that DeepReduce can achieve an average of 98.1% of the original accuracy (the test accuracy achieved by the whole testing data). In the worst case, the accuracy achieved by the selected data is still over 94.3% (0.8316/0.8815) of the original one. The results show that our approach can save more than 95.5% (1 - 4.5%) testing cost on average. Besides, the selected data of DeepReduce can achieve a similar test accuracy as the whole testing data achieves, which indicates that DeepReduce is effective in estimating the performance of DL models.

When compared with other approaches, DeepReduce outperforms the random approach. From the table, we find that when using the same termination criteria, the random approach performs much worse than DeepReduce as it requires much more testing data in order to obtain a representative reduction result. Notice that the amount of selected data obtained by the random approach ranges from 2,536 (25.4%) to 4,903 (49.0%), with an average of 3,291 (32.9%). That is, DeepReduce outperforms the random approach 8.0X on average in size, while providing similar performance estimations.

In order to compare the state-of-the-art approach with our approach, we implement $BOT_v$ by using the same termination criteria as the one used in DeepReduce. From the table, we can find that both DeepReduce and $BOT_v$ perform well in input reduction. We also find that when using the same termination criteria, $BOT_v$ selects 11.8% more testing data compared with DeepReduce. In extreme cases, $BOT_v$ selects 44.6% more testing data for LeNet5. In summary, considering the amount of the selected data, our approach outperforms $BOT_v$ on five models except VGG19.

Note that $BOT_v$ is a random sampling approach with a carefully designed strategy, and it tends to perform differently in different executions. Thus, it is not sure whether $BOT_v$ can achieve the average performance in practice every time it is used. As an example, we show more results on NIN to further illustrate the instability of $BOT_v$. On NIN, the test accuracy achieved by the data selected by $BOT_v$ ranges from 0.7925 to 0.88, and the number of selected data instances ranges from 525 to 605. In comparison, DeepReduce is more stable than $BOT_v$ as it does not use random sampling. Furthermore, $BOT_v$ only considers output distribution in reducing testing data, and does not guarantee testing adequacy for testing DL models. In contrast, our approach ensures testing adequacy by selecting data that can cover all neurons covered by the whole testing data. Thus, we believe that DeepReduce is more practical than $BOT_v$ as it is more stable and reliable.

To sum up, DeepReduce can obtain accurate estimation of the performance of DL models using less testing data. Furthermore, it outperforms the random approach and is more practical than $BOT_v$.

### 4.3.2 RESULTS FOR RQ2

We investigate the performance of our approach under different values of parameters (i.e., $\beta$ and $\alpha$). We set $\alpha$ in the termination criteria to 0.05, 0.01, 0.005, and 0.001, and $\beta$ in the coverage criteria to 0.25, 0.5, and 0.75. We present the results on CIFAR-10 in Table 3 and the results on MNIST are available online (DeepReduce, 2019). Note that our approach is a two-phase reduction algorithm, therefore we also present the number of selected data instances in the first phase (by using HGS) in Column "HGS", as it is interesting to known how much testing data is needed for achieving the same testing adequacy as the whole testing data does.

| Subjects | Coverage | HGS | KL < 0.05 | | KL < 0.01 | | KL < 0.005 | | KL < 0.001 | |
|---|---|---|---|---|---|---|---|---|---|---|
| | | | Acc | Size | Acc | Size | Acc | Size | Acc | Size |
| NIN (0.8815) | NC0.25 | 15 | 0.7602 | 196 | 0.8380 | 358 | 0.8597 | 506 | 0.8633 | 1405 |
| | NC0.5 | 68 | 0.6985 | 199 | 0.8011 | 357 | 0.8316 | 487 | 0.8478 | 1334 |
| | NC0.75 | 186 | 0.7764 | 246 | 0.8136 | 381 | 0.8356 | 517 | 0.8663 | 1436 |
| VGG19 (0.9346) | NC0.25 | 24 | 0.9062 | 128 | 0.9329 | 313 | 0.9382 | 437 | 0.9340 | 1000 |
| | NC0.5 | 108 | 0.8898 | 236 | 0.9279 | 416 | 0.9312 | 552 | 0.9381 | 1115 |
| | NC0.75 | 635 | 0.8333 | 1218 | 0.8896 | 2020 | 0.9035 | 2425 | 0.9183 | 3440 |
| ResNet (0.9220) | NC0.25 | 8 | 0.7967 | 305 | 0.8710 | 589 | 0.8835 | 824 | 0.9058 | 2059 |
| | NC0.5 | 23 | 0.7922 | 308 | 0.8639 | 617 | 0.8779 | 827 | 0.9039 | 2071 |
| | NC0.75 | 98 | 0.7832 | 286 | 0.8592 | 568 | 0.8793 | 787 | 0.9022 | 2005 |

Table 3: Experiment results using different parameters on CIFAR-10

The results show that only a small number of data instances is required in order to achieve the same coverage as the whole testing data achieves, under different coverage criteria and termination criteria. The results also show that, at the first phase of our approach (HGS), more data is required when a higher $\beta$ is used. This is reasonable as the neuron coverage is harder to achieve under a higher activation threshold. The approximation of output distribution at the second phase selects more data instance than HGS. That is, testing adequacy can be achieved by selecting a smaller amount of data, while more data are needed in order to achieve the similar output distribution. In terms of test accuracy, in most cases, the results achieved when setting $\beta$ to different values are similar when the same $\alpha$ is used in the termination criteria. When the termination criterion gets stricter, the accuracy becomes higher. In our experiments, we set $\alpha = 0.005$ and $\beta = 0.5$ as default settings.

### 4.3.3 RESULTS FOR RQ3

In order to evaluate our reduction approach in regression scenarios, we apply seven changes to the studied DL models, re-train the modified models, and then test them using the data selected by DeepReduce (as described in RQ1). Table 4 shows the average testing results on the NIN model. The whole results are available online (DeepReduce, 2019). In the table, "Actual" presents the actual accuracy achieved by the whole testing data, "DeepReduce" presents the accuracy achieved by the testing data selected by DeepReduce, and "$\Delta$" presents the difference between the two. The experiment results show that DeepReduce performs well in estimating the performance of DL models even when they are modified. The average of $\Delta$ ranges from 0.0307 to 0.0424, indicating that the difference between the estimated and actual test accuracy is small.

| Model | Metrics | Modifications | | | | | | |
|---|---|---|---|---|---|---|---|---|
| | | Structural Changes | | | | Non-Structural Changes | | |
| | | ADL | DEL | ADN | DEN | LR | MO | DR |
| NIN | Actual | 0.8816 | 0.8734 | 0.8857 | 0.8808 | 0.8480 | 0.8700 | 0.8819 |
| | DeepReduce | 0.8490 | 0.8391 | 0.8545 | 0.8447 | 0.8056 | 0.8393 | 0.8434 |
| | $\Delta$ | 0.0326 | 0.0343 | 0.0313 | 0.0360 | 0.0424 | 0.0307 | 0.0385 |

Table 4: Effectiveness of our approach in regression scenarios

## 5 CONCLUSION

In this paper, we propose DeepReduce, a software engineering approach for cost-effective testing of DL models. Our approach can accurately estimate the performance of a DL model using a small subset of testing data, by considering three different objectives: efficiency, completeness, and effectiveness. We have evaluated DeepReduce on six DL models and two datasets. On average, DeepReduce can reduce the whole testing data to 4.6%. The results confirm that the proposed approach can significantly reduce DL testing cost by reducing the amount of testing data required for testing a DL model.

The implementations of DeepReduce and the data used in our experiments are available at: https://github.com/DeepReduce/DeepReduce .

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
