# OpenReview forum: "Cost-Effective Testing of a Deep Learning Model through Input Reduction"
_ICLR.cc/2020/Conference — Reject_

### Official Review · AnonReviewer1 · 2019-10-21
**Official Blind Review #1**

**Rating:** 8

**Review:**

The paper presents a new approach to create subsets of the testing examples that are representative of the entire test set so that the model can be tested quickly during the training and leaving the check on the full test set only at the end to validate its validity.

The key idea is to create the smaller possible subset with the same or similar coverage (in the paper the neurons coverage is considered) and output distribution, maintaining the difference below a (small) threshold.

In particular, the output distribution is approximated by dividing the output range of each neuron into K intervals. In this way, an estimate of the output distribution is considered during the extraction of the representative subset of the testing data.

The whole process is divided into two phases, the first is to create a first subset using HGS, the second refines this subset in order to achieve the desired precision in the output distribution.

The paper also presents a good experimental campaign that shows good performances.

The paper is really well written and enjoyable, clear in its description and in the objectives it aims to achieve. To improve the readability a bit further, I would suggest trying to move equation 1 after or close to its reference, or at least to describe before it what KL is.
Moreover, a similar problem is present for TI, which is introduced in Algorithm 1 and described later. I would add, for example in the input section of the algorithm, a sentence stating that TI is used in getCandidate and described later.

The first sentence on page 5 seems to be incomplete as it is written. I would suggest rephrasing the sentence.

In Algorithm 2, about the two "for" cycles for i in 1,m and foreach k in 2,K, I suggest unifying them and use the same cycle (only for reasons of readability). Moreover, I think that using braces instead of parentheses would be more correct in these cycles.

It is interesting to note that in the first line for VGG19, last column of Table 3, the accuracy is lower with KL < 0.001 than with KL < 0.005. I would expect monotonicity here. Do the authors have any idea of the reasons for this?

Finally, it would be interesting to know the runtime to obtain the subsets of the test data of Table 2 required by the considered systems.

Typos:
On page 6, penultimate paragraph, the word "the" is repeated twice in the parentheses.
On page 8, at the beginning of the first sentence after Table 3, there is a comma that seems to be useless after the word "that".
On page 8, "When the termination crite gets stricter", crite should be corrected in criterion.
There is a typo in the README of the github project linked in the paper: "coveraeg data" instead of "coverage data".

**Experience Assessment:**

I have read many papers in this area.

**Review Assessment: Checking Correctness Of Derivations And Theory:**

N/A

**Review Assessment: Checking Correctness Of Experiments:**

I carefully checked the experiments.

**Review Assessment: Thoroughness In Paper Reading:**

I read the paper at least twice and used my best judgement in assessing the paper.

---

> ### Author Response · Authors · 2019-11-11
> **Author Response to Reviewer #1**
>
> We thank the reviewer for your helpful and encouraging comments. We have addressed the comments and revised the paper.  Please see the detailed replies below:
>
> 1. ’'...the accuracy is lower with KL < 0.001 than with KL < 0.005. I would expect monotonicity here.'
> Response: We agree that there is monotonicity among the results with different KL values. The monotonicity can be generally observed in Table 3. The only exception is on VGG19 when KL<0.005 and NC=0.25. This is because our approach is a greedy heuristic, the monotonicity may not be very strict when the absolute difference in accuracy is small. For VGG19, when KL<0.005 and NC=0.25, the absolute difference in accuracy is less than 0.004, which could be caused by just one or two test data points.
>
>
> 2. About the runtime to obtain the subsets of the test data of Table 2
> Response: The runtime to obtain the subsets varies for different models. For example, the runtime for three LeNet models ranges from 8.93 to 16.27 seconds. More details will be given on our website.
>
>
> 3. About the presentation and typos:
> Response: Thanks for your suggestions! We have improved the presentation of the paper and fixed the typos.

---

### Official Review · AnonReviewer2 · 2019-10-23
**Official Blind Review #2**

**Rating:** 6

**Review:**

The paper develops methods to reduce the test data size while maintaining the coverage and the effectiveness on large test data. The paper proposes  a two-phase reduction approach to select representative samples based on heuristics. Extensive experiments have shown the proposed methods can reduce 95% test samples while still obtaining similar measurements.

The paper targets a very important problem in practice. Effectively selecting small, representative test sets can save many computational resources and greatly accelerate the research and development. Although the developed technique is quite simple, they are meaningful in practice. Overall, the work can be much improved if a theoretical framework is proposed.



**Experience Assessment:**

I do not know much about this area.

**Review Assessment: Checking Correctness Of Derivations And Theory:**

I assessed the sensibility of the derivations and theory.

**Review Assessment: Checking Correctness Of Experiments:**

I assessed the sensibility of the experiments.

**Review Assessment: Thoroughness In Paper Reading:**

I read the paper at least twice and used my best judgement in assessing the paper.

---

> ### Author Response · Authors · 2019-11-11
> **Author Response to Reviewer #2**
>
> We thank the reviewer for your helpful and encouraging comments. Please see the detailed replies below:
>
> 1. Regarding the comment on theoretical framework.
> Response: Thanks for your valuable suggestion! As you point out, our paper investigates the input reduction problem from a  practical perspective. It would be interesting to investigate the theoretical framework in our future work.

---

### Official Review · AnonReviewer3 · 2019-10-23
**Official Blind Review #3**

**Rating:** 3

**Review:**

This work tries to build a sub-pile of the test data to save the testing time with minimum effect on the test adequacy and the output distribution. In this paper, the work is done by adding a test-sample search algorithm on top of the HGS algorithm to balance the output distribution.

However, the novelty of the proposed work is limited, and there is no evidence to show the proposed algorithms can be applied to other related works. Furthermore, the result does not present a strong success: the error of output distribution is much worse than the compared work.

Other comments to the proposed manuscript are:

1. In Definition 1 the authors declare that the goal is to satisfy f(T,M)=f(T’M) and g(T,M)=g(T’M), and then in the following paragraphs they change it to f(T,M)≈f(T’M) and g(T,M)≈g(T’M) with no justification. More explanation is needed.

2. In Table 2, the authors try to compare the output distribution. To better demonstrate the change between raw testing set and proposed subset, I think that it could be better to present the metrics of distribution or the accuracy of each class instead.


**Experience Assessment:**

I have published one or two papers in this area.

**Review Assessment: Checking Correctness Of Derivations And Theory:**

I assessed the sensibility of the derivations and theory.

**Review Assessment: Checking Correctness Of Experiments:**

I assessed the sensibility of the experiments.

**Review Assessment: Thoroughness In Paper Reading:**

I read the paper at least twice and used my best judgement in assessing the paper.

---

> ### Author Response · Authors · 2019-11-11
> **Author Response to Reviewer #3 : part 1**
>
> We thank the reviewer for your valuable comments. The details of our response are as follows:
>
> 1. Regarding the comment on the novelty of the proposed work.
> Response: We believe that our work is significant and novel. The novelty can be summarized as follows:
> (1) Our work is the first work to define multi-objective input reduction problem in DL testing, and the first work to reduce the cost of DL testing by satisfying three objectives (efficiency, completeness, and effectiveness). Related work such as BOT focuses on single-objective (effectiveness) and does not ensure testing completeness.
>
> (2) We propose a two-phase reduction algorithm, which is a new approach to input reduction. The two-phase algorithm combines three objectives by using the HGS algorithm and a carefully designed heuristic. Our approach is a meaningful approach in practice, which is also accepted by another reviewer (#2).
>
> (3) An extensive evaluation of the proposed approach, including non-regression and regression scenarios. The latter is a more practical scenario in the development process. Our approach can work well in both scenarios.
>
> 2. Regarding the applicability of the proposed algorithms to other related works
> Response: Our algorithm is general and can be easily adapted to other related work (e.g., other criteria and DL models). In our work, we use neuron coverage and the outputs of the last hidden layer as the two inputs. However, our algorithm is not specific to the neuron coverage criterion, and can support various other coverage criteria. That is, if the coverage of a DL model can be obtained, the first phase (HGS) of our algorithm can also be applied to it. Besides, for most of the DL models, the outputs of the last hidden layer (or the last few hidden layers) are numerical values, indicating that the latter phase of our algorithm can be applied to other DL models too. To sum up, our algorithm can be easily applied to other work as well.
>
> Note that there are some other related works targeting the cost problem in DL (e.g., those described in Section 2). Our approach is proposed to solve the input reduction problem in DL testing, and we do not claim that our approach can be applied to the work described in Section 2, as these work are quite different from our work. However, our approach and these work can complement each other in order to reduce DL costs.

---

> > ### Author Response · Authors · 2019-11-11
> > **Author Response to Reviewer #3 : part 2**
> >
> > (continued reponses)
> >
> > 3. ”Furthermore, the result does not present a strong success: the error of output distribution is much worse than the compared work.”
> > Response: Our approach is significantly better than the compared work and presents a strong success. Our reasons are as follows:
> > (1)  The compared work is BOT (Li et al., 2019), which is the state-of-the-art approach in DL testing. As described in Section 4.2 “Experimental Design”, different from DeepReduce, BOT requires users to specify the number of data instances to be selected, whose optimal value is unknown to users before using BOT. To enable comparison, we actually implemented a variant of BOT, called BOTv, by changing the termination criteria to the one used in DeepReduce. That is, the state-of-the-art approach BOT itself cannot be directly used to compare with our work. The results of BOTv shown in Table 2 already incorporate the knowledge (i.e., the KL value) of our work.
> >
> > (2) Note that our work tries to obtain a representative subset of testing data, thus we expect the accuracy of the selected subset to be close to the original one. Our accuracy (the error of output distribution) is only slightly worse than that of BOTv. The results in Table 2 show that the absolute differences between our accuracy (the error of output distribution) and the original one are only slightly worse than those of BOTv (ranging from 0.0018 to 0.0255, with an average of 0.0105). In four out of six studied models, the differences are less than 0.01 and only in one studied model the difference is more than 0.02 (0.0255).
> >
> > (3) Although BOTv performs slightly better than DeepReduce in accuracy,  it does not consider satisfying structural coverage in data selection. Therefore, testing completeness cannot be ensured by BOTv. For example,  the average neuron coverage (when the threshold of NC is 0.75) achieved by BOTv is 92.7% of the coverage achieved by the original testing set. However, our proposed approach can ensure both testing completeness (always achieving 100% coverage) and testing effectiveness.
> >
> >
> > (4) As described in Section 4.3.1, our approach is more stable than BOTv, which adopts a random sampling strategy. We cannot guarantee that the average effectiveness can be achieved every time we use BOTv. For example, on NIN, the accuracy achieved by BOTv ranges from 0.7925 to 0.88, which varies a lot. In comparison, DeepReduce is more stable than BOTv as it does not use random sampling. Therefore, our approach is more useful in practice.
> >
> > (5) Although our approach is slightly worse than BOTv in terms of overall accuracy, it is better than BOTv in terms of size, indicating that our approach can save more testing costs in DL testing.
> >
> > In summary, we believe that our approach is significantly better than the compared work (BOT/BOTv) and presents a strong success.  Reviewer #1 also confirms that the performance of our approach is good.
> >
> >
> > 4. About the change of definition to f(T,M)≈f(T’M) and g(T,M)≈g(T’M) without any explanation.
> > Response: According to the third paragraph of Section 3.1, we do not change the sub-goal f(T,M)=f(T’,M) to f(T,M)≈f(T’,M). We only replace one sub-goal g(T,M)=g(T',M) with g(T,M)≈g(T',M). This is because the defined input reduction problem (with the two sub-goals) is NP-complete. We relax the requirement on the output distribution g(T,M) so that our algorithm is more likely to produce a subset of input in an acceptable time. A brief explanation is given in this paragraph, including the representative meaning of '≈'. We will add more explanations in this paragraph.
> >
> >
> > 5. Regarding the comment on “the metrics of distribution or the accuracy of each class”
> > Response: Thanks for your suggestion! In Table 2, we showed the overall accuracy of all classes. The difference between the proposed subset and the original testing set (|accuracy of proposed subset - accuracy of ran testing set|) is 0.0284 on average (ranging from 0.0008 to 0.1057), which is similar to the observation on the overall accuracy of all classes. The results indicate that the proposed subset can be used to represent the raw testing data.

---

### Decision · Program_Chairs · 2019-12-19

**Decision:**

Reject

**Comment:**

This paper presents a method which creates a representative subset of testing examples so that the model can be tested quickly during the training. The procedure makes use of the famous HGS selection algorithm which identifies and then eliminates the redundant and obsolete test cases based on two criteria: (1) structural coverage as measured by the number of neurons activated beyond a certain threshold, and (2) distribution mismatch (as measured by KL divergence) of the last layer activations. The algorithm has two-phases: (1) a greedy subset selection based on the coverage, and (2) an iterative phase were additional test examples are added until the KL divergence (as defined above) falls below some threshold.
This approach is incremental in nature -- the resulting multi-objective optimisation problem is not a significant improvement over BOT. After the discussion phase, we believe that the advantages over BOT were not clearly demonstrated and that the main drawback of BOT (requiring the number of samples) is not hindering practical applications. Finally, the empirical evaluation is performed on very small data sets and I do not see an efficient way to apply it to larger data sets where this reduction could be significant. Hence, I will recommend the rejection of this paper. To merit acceptance to ICLR the authors need to provide a cleaner presentation (especially of the algorithms), with a focus on the incremental improvements over BOT, an empirical analysis on larger datasets, and a detailed look into the computational aspects of the proposed approach.